# Metformin Protects Rat Skeletal Muscle from Physical Exercise-Induced Injury

**DOI:** 10.3390/biomedicines11092334

**Published:** 2023-08-22

**Authors:** Giuliana Abbadessa, Eleonora Maniscalco, Loredana Grasso, Jasmin Popara, Federica Di Scipio, Francesco Franco, Daniele Mancardi, Fabio Pigozzi, Paolo Borrione, Giovanni Nicolao Berta, Silvia Racca

**Affiliations:** 1Department of Clinical and Biological Sciences, University of Turin, 10043 Orbassano, Italy; eleonora.maniscalco@unito.it (E.M.); loredana.grasso@unito.it (L.G.); jasmin.popara@uksh.de (J.P.); federica.discipio@unito.it (F.D.S.); francesco.franco@unito.it (F.F.); daniele.mancardi@unito.it (D.M.); silvia.racca@unito.it (S.R.); 2Department of Movement, Human and Health Sciences, University of Rome “Foro Italico”, 00135 Rome, Italy; fabio.pigozzi@uniroma4.it (F.P.); paolo.borrione@uniroma4.it (P.B.)

**Keywords:** skeletal muscle, metformin, training, physical performance, muscle adaptation

## Abstract

Metformin (Met) is a drug commonly prescribed in type 2 diabetes mellitus. Its efficacy is due to the suppression of hepatic gluconeogenesis, enhancement of peripheral glucose uptake and lower glucose absorption by the intestine. Recent studies have reported Met efficacy in other clinical applications, such as age-related diseases. Despite the wide clinical use of Met, its mechanism of action on muscle and its effect on muscle performance are unclear. We investigated the effects of Met combined with training on physical performance (PP) in healthy rats receiving Met for 8 weeks while undergoing daily moderate exercise. We evaluated the following: PP through graded endurance exercise test performed before the beginning of the training protocol and 48 h before the end of the training period; blood ALT, AST, LDH and CK–MB levels in order to address muscle damage; and several blood and muscle myokines and the expression of factors believed to be involved in muscle adaptation to exercise. Our data demonstrate that Met does not improve the positive effects of exercise on performance, although it protects myocytes from exercise-induced damage. Moreover, given that Met positively affects exercise-induced muscle adaptation, our data support the idea of the therapeutic application of Met when muscle function and structure are compromised.

## 1. Introduction

Metformin (Met) is one of the most prescribed drugs for the treatment of type II diabetes mellitus due to its ability to reduce blood glucose by decreasing glucose production in the liver and to increase peripheral glucose disposal in adults with and without type II diabetes mellitus [1,2,3].

More recently, there has also emerged a growing interest in using Met to treat ageing and to delay the onset of several age-related diseases [4].

The molecular mechanisms that underlie Met action still need elucidation, although some of them have been identified. Met has been shown to inhibit mitochondrial respiration at complex I of the electron transport system [5] establishing an energetic stress in skeletal muscle and activating adenosine monophosphate-activated protein kinase (AMPK) [6]. AMPK is considered a cellular energy sensor and, in response to energetic stress, it increases energy-producing features such as skeletal muscle mitochondrial biogenesis and insulin-stimulated glucose uptake. In addition, activated AMPK increases hepatic and muscle fatty acid oxidation and decreases hepatic glucose, cholesterol, and triglyceride production. Additionally, Met has been demonstrated to decrease skeletal muscle mitochondrial reactive oxygen species emission, and consequently improve insulin sensitivity [7].

From a clinical point of view, Met and the other oral antidiabetic drugs are able to stimulate, amplify, or modulate insulin secretion or its biological effects. Although insulin is inserted in the list of the substances and methods banned by the World Anti-doping Agency (WADA) [8], and its consumption is not allowed for healthy athletes (except for those affected by type 2 diabetes), oral antidiabetic drugs, including Met, are not included in this list. Furthermore, it is unclear whether these drugs can be considered an alternative to insulin for those athletes who want to take advantage of the hormone action on skeletal muscle to enhance their physical performance (PP).

Physical activity is one of the most efficient and accessible therapeutic interventions to delay the process of skeletal muscle involution due to ageing (sarcopenia), to ameliorate the symptoms of various chronic diseases and to give rise to pleiotropic health benefits. Though the mechanisms underlying the plastic changes of skeletal muscle after exercise are far from being fully understood, several important regulators have been identified. For example, AMPK has been strongly implicated in the exercise-induced molecular signaling involved in the regulation of skeletal muscle metabolism and in the critical adaptive processes for enhanced muscle contractile and metabolic functions in the long run [9]. Moreover, AMPK has been implicated in the promotion of mitochondrial remodeling in skeletal muscle, a classic adaptation to the energetic demands of aerobic exercise.

Within skeletal muscle, aerobic exercise training and Met independently have positive effects, which are thought to be partially mediated by AMPK stimulation. Additionally, both Met and endurance training enhance the muscle satellite cell pool [10,11]. The decline of satellite cells with age strongly compromises the regenerative capacity of skeletal muscle. Thus, satellite cells may be the major target of the sarcopenic ageing process [12].

Several authors have evaluated the effects of the summation of exercise and Met therapy on glucose metabolism [13,14,15], as well as on microvascular function and cardiovascular indices [16]. Met’s impact on exercise capacity has been evaluated principally in unhealthy subjects [17,18,19] but little is known about the effects of Met in healthy subjects [20] and especially in reference to elite athletic performances.

Based on this background, our study aims to:

(i) evaluate if Met, which is not a doping substance per se, is able to enhance muscular performances and should then be considered as a doping substance;

(ii) analyze the effects of Met on the principal regulators of exercise-induced skeletal muscle adaptation;

(iii) clarify how Met administration combined with an exercise program can impact on the beneficial effects of exercise on muscle tissue.

Because the inclusion of Met in the WADA list of prohibited substances is under discussion, providing precise knowledge on how muscle performance responds to its administration is crucial. To date, there is great controversy around the question of whether Met treatment combined with exercise training improves or nullifies the benefits provided by physical activity on muscle performance. This research may be useful to clarify the role of Met in this context.

## 2. Materials and Methods

### 2.1. Animals and Pharmacological Treatment

Thirty-two months old male Wistar rats weighing ~250 g (Harlan Laboratories, San Pietro al Natisone, Udine, Italy) were housed in the department’s animal facility with access to food and water ad libitum and maintained in a controlled environment (12:12 light–dark cycle, room temperature 20–24 °C, and 50–60% humidity). Animals were allowed 1 week of acclimatization before the experiment began. All experimental procedures were conducted according to the guidelines of the Declaration of Helsinki and approved by the Italian Ministry of the Health, protocol code 1051/2015-PR, date of approval 2 October 2015.

Animals were randomly divided into the following groups:-Control (n = 12): rats submitted to graded endurance exercise test (GEET) as described below, and placed on a stationary treadmill daily for 5 min. Six animals received normal drinking water (C−) and the remaining six animals received Met (metformin hydrochloride, 98%—Bosche Scientific) dissolved in drinking water at a dose of 250 mg/kg body weight until sacrifice (C+).-Exercised (n = 12): rats submitted to training on treadmill daily for 8 weeks and to GEET. Six animals were given normal drinking water (EX−) and the remaining six rats received Met dissolved in drinking water at a dose of 250 mg/kg body weight until sacrifice (EX+).-Sedentary (n = 6): rats not submitted to any type of exercise and placed at rest on the treadmill daily for 5 min (SED). This group was included in the study design to check for possible effects of GEET. We showed the results obtained in this group only when they differed significantly from C−.

The Met dose was based upon previous reports [20] and its addition does not influence water consumption. Water and Met were replaced daily, and the dose adjusted to weight gain weekly.

Body weight was monitored twice a week throughout the experimental period. The body mass of animals subjected to running was measured before training. Water intake was recorded every day while food intake was measured two days a week.

Blood was collected and then animals were sacrificed 24 h after the last training session with an excess of anesthetic. The gastrocnemius muscle was excised, weighed, and split into three parts: one fixed in 4% paraformaldehyde, another used to extract the mitochondria and the last frozen in liquid N_2_ and stored at −80 °C until processing and analysis.

### 2.2. Exercise Training Protocol

Rats were exercised on a five-lane motorized rodent treadmill (Biological Instruments, Besozzo (VA), Italy). The training protocol consisted of daily treadmill run sessions representing moderate exercise [21,22,23,24]. The protocol was repeated for a total of 8 weeks, 5 days per week, at the same time each day. The treadmill was equipped with a mild electric shock at the end of each lane to motivate rats and to ensure a continuous exercise.

The intensity of training was kept low during the first week to habituate rats to the treadmill. This phase consisted of a run at a speed of 25 cm/s for 5 min without slope; thereafter, the running speed and duration were progressively increased until, after 4 weeks, the rats ran continuously for 60 min/day, 34 cm/s without slope. Then, during the following four weeks, the speed and duration remained constant (Table 1).

### 2.3. GEET

All rats, except those of the SED group, underwent GEET (Figure 1) in two different times of the experimental period: the day before the beginning of the training protocol and the 54th day of the training period (i.e., 48 h before the sacrifice of the animals).

This test consisted of a progressive exercise in which each rat initially ran at a speed of 8 cm/s, up a 0% slope, for 3 min. Thereafter, the speed was increased by 8 cm/s every 3 min until the rat was unable to keep the pace with the treadmill belt despite the application of an electrical stimulus [25].

### 2.4. Haematochemical Parameters

Blood from each animal was collected in a stopper tube (without anticoagulants) and centrifuged: serum samples were then immediately frozen for further biochemical analyses. Abbott specific kits for automated chemistry analyzer (Architect C8000 Plus, Abbott, Sesto San Giovanni (MI) Italy) were used for the evaluation of alanine aminotransferase (ALT), aspartate aminotransferase (AST), creatinine, glucose, triglycerides, urea, total cholesterol, high density lipoprotein (HDL), C-reactive protein (PCR), creatine kinase MB (CK–MB) and lactate dehydrogenase (LDH), according to the manufacturer’s instructions.

### 2.5. Myokines Assay

Millipore’s Milliplex^®^ Map-Rat Myokine Magnetic Bead Panel (EMD Millipore, Merk Life Science S.r.l, Milano, Italy) was used in serum and muscle total lysate for the simultaneous quantification of the following cytokines: brain-derived neurotrophic factor (BDNF), erythropoietin (EPO), interleukin-15 (IL-15), fibroblast growth factor 21 (FGF21), fractalkine (FKN), interleukin-6 (IL-6), follistatin-like 1 (FSTL-1), myostatin (MSTN/GDF8), irisin, leukaemia inhibitory factor (LIF), osteocrin, and secreted protein acid and cysteine rich (SPARC).

### 2.6. Total Muscle Lysate and Mitochondrial Fraction

Muscle samples were homogenized in an RIPA lysis buffer (150 mM NaCl, 1.0% IGEPAL^®^ CA-630, 0.5% sodium deoxycholate, 0.1% SDS, 50 mM Tris, Sigma-Aldrich, Merk Life Science S.r.l, Milano, Italy) supplemented with protease inhibitor cocktail (Cell Signaling, Euroclone S.p.A., Pero (MI) Italy), by using a Potter homogenizer and then centrifuged at 16,100× *g* for 30 min at 4 °C. The supernatant was collected and stored at −80 °C for further examinations. Mitochondrial fraction was extracted as described by Penna et al. [26] starting from fresh muscle samples. Protein concentrations of muscle total lysate and mitochondrial extract were determined by Bradford colorimetric assay (Bio-Rad, Segrate (MI) Italy) [27].

### 2.7. Western Blot (WB)

Total protein extract from muscle lysates (40 µg) and mitochondrial fraction (2.5 µg) were separated by SDS–PAGE. Gels were transferred to membranes, saturated with blocking solution (5% milk and 0.1% Tween-20 in PBS), and incubated with the suitable primary antibody (Appendix A) overnight at 4 °C.

Membranes were then rinsed three times and incubated with the appropriate concentrations of secondary antibody conjugated with horseradish peroxidase for 1 h at room temperature. Blots were developed with Clarity Western ECL Substrate (Bio-Rad) using ChemiDocTM Touch Image System (Bio-Rad). Densitometric analysis was performed using ImageLab 6.1 Software. Non-phosphorylated proteins were normalized to glyceraldehyde-3–phosphate dehydrogenase (GAPDH) or vinculin. Phosphorylation level is presented as the ratio between phosphorylated and total protein.

All original Western blot images are presented in Appendix A (see Appendix A).

### 2.8. Immunohistochemical (IHC) Analysis

Muscles samples were fixed in 4% paraformaldehyde phosphate-buffered saline (PBS) for 3 h, washed in PBS and embedded in paraffin after dehydration with ascending ethanol passages (50, 70, 80, 95, 100%) followed by diaphanization in Bioclear (Bio-Optica, Milano, Italy). *Gastrocnemii* were sectioned (7 µm thick) using an RM2135 microtome (Leica Microsystems) and placed on slides. They were then deparaffinized and rehydrated with decreasing ethanol passages.

Immunohistochemistry staining was performed using IHC Select^®^ HRP/DAB (Merck Millipore, Burlington, MA, USA) according to the manufacturer’s instructions. Briefly, slides were treated with 0.1% trypsin solution to recover tissue antigenicity. Then, 3% hydrogen peroxide solution was used to block endogenous peroxidase activity. After an incubation of 5 min in blocking reagent, slides were incubated overnight with primary antibodies (Appendix A) at 4 °C. The next day, the secondary antibody included in the kit was added to the slides for 10 min, followed by a first incubation with streptavidin HRP (10 min) and then a second incubation with the chromogen reagent (8 min).

The sections were dehydrated with ascending ethanol passages and mounted in dibutylphthalate polystyrene xylene (DPX), (BDH) [28].

### 2.9. Statistical Analysis

All data are presented as mean values ± SD of at least three independent experiments. Statistical analyses were performed with *t*-test or two-way ANOVA, followed by Tukey multiple comparison test by using GraphPad Prism 8.0 software; *p*-values ≤ 0.05 were considered statistically significant.

## 3. Results

### 3.1. Food Consumption, Body Weight Increase and Muscle Weight/Body Weight Ratio

As shown in Figure 2, no significant differences in food consumption and body weight increase emerged among groups.

As displayed in Figure 3, Met administration increased the muscle weight/body weight ratio (*F* = 11.25, *p* = 0.004). Tukey’s post-hoc test confirmed significant differences in C+ and EX+ groups compared with C−.

### 3.2. Evaluation of PP

To evaluate how training and Met administration affected rats’ PP, animals underwent the GEET at the beginning and at the end of the training period. Comparison between the two GEET showed that physical training led to a better performance, in terms of covered distance compared with untrained animals (C− and C+). A significant increase was evidenced in EX− and EX+ compared with C− and C+ respectively. An increased performance was observed in C+ compared with C−, although it was not significant. The improvement in PP due to training was slightly improved over that induced by drug administration in EX+ (Figure 4).

### 3.3. Haematochemical Parameters

Table 2 reports the results obtained from the blood analysis. We observed differences in ALT, AST, LDH and CK–MB values between SED and C−. Although each of these was higher in C− than in SED, the difference was statistically significant only for ALT (*t*-test, *p* = 0.031).

The comparison among control and exercised groups, treated or not, showed that training decreased ALT (*F* = 14.10, *p* = 0.0021) (Figure 5a) and CK–MB (*F* = 6.53, *p* = 0.021) levels (Figure 5d); Met administration reduced AST (*F* = 7.46, *p* = 0.015) and LDH (*F* = 5.94, *p* = 0.027) levels (Figure 5b,c).

### 3.4. Myokines in Serum and in Total Muscle Lysate

A panel of 12 pertinent myokines was evaluated in samples of serum and muscle lysate. As regards the myokines levels in serum, significant differences among groups emerged in irisin, FGF21, osteocrin and SPARC levels. Met administration accounted for the increase in irisin (*F* = 4.50, *p* = 0.050) and FGF21 (*F* = 20.21, *p* = 0.0004). Nevertheless, Tukey’s post-hoc test evidenced a meaningful difference only in FGF21 levels in C+ vs. C− and in EX+ vs. EX− (Figure 6a,b).

Osteocrin serum levels were modulated by Met administration (*F* = 7.796, *p* = 0.013), training (*F* = 8.792, *p* = 0.009), and interaction (*F* = 11.31, *p* = 0.004). Tukey’s post-hoc test confirmed significantly lower levels in C+, EX− and EX+ compared with C− (Figure 6c).

As regards SPARC data, drug administration led to a serum increase of this myokine (*F* = 7.072, *p* = 0.017) (Figure 6d).

In muscle lysate samples significant differences among groups emerged only for FKN. Statistical analysis reported significant effects for Met administration (*F* = 68.66, *p* < 0.0001) and training (*F* = 9.907, *p* = 0.004) but not for interaction. Tukey’s post-hoc test confirmed a significant decrease in both C+ and EX− versus C−, and in EX+ versus EX− (Figure 6e).

### 3.5. WB Analysis

AMPK and some factors involved directly and indirectly in its pathway, such as acetyl-CoA carboxylase beta (ACCß), peroxisome proliferator-activated receptor gamma coactivator-1 alpha (PGC-1α), glycogen synthase kinase 3 beta (GSK3ß), protein kinase B (AKT) and the mammalian target of rapamycin (mTOR) were analyzed in muscle samples.

A significant increase of AMPK activity, expressed as a phospho(p)-AMPK/AMPK ratio, was observed in C− with respect to SED (*t*-test, *p* < 0.0001) (Figure 7a). Met administration significantly increased the p-AMPK/AMPK ratio in C+ and EX+ in comparison with C− and EX−, respectively. Although training alone did not modify AMPK activation, when it was combined with Met administration it blunted the p-AMPK/AMPK ratio increase induced by Met (Figure 7b).

Regarding ACCß, which is inactivated when phosphorylated, the p-ACC/ACCß ratio increased in trained rats, both in EX− and in EX+, compared with the respective control groups (C− and C+). Furthermore, the increase observed in EX+ was significantly higher than in EX− (Figure 8).

PGC-1α expression in the mitochondrial fraction was increased by Met administration and even further by training with respect to C−. Nevertheless, PGC-1α expression reached a higher expression level when Met administration was combined with training showing a synergistic effect (Figure 9a). To confirm these data, we evaluated the expression of cytochrome C (Cyt C) that is induced by PGC-1α, and we obtained the same pattern observed for PGC-1α (Figure 9b).

AKT phosphorylation increased significantly in EX+ compared with EX− and C+ (Figure 10a).

GSK3ß is a kinase which is phosphorylated when inactivated. The p-GSK3ß/GSK3ß ratio was higher in C+ and EX− versus C−, with a synergistic effect of Met and training in EX+ (Figure 10b).

mTOR phosphorylation was significantly increased by Met administration (C+) and training (EX−) compared with C−. Moreover, a significant increase of p-mTOR/mTOR ratio was evident in EX+ compared with EX− and C+ (Figure 11a) without interaction.

Representative images of ribosomal protein S6 kinase (p70S6K) and p-p70S6K are shown in Figure 11b. Although it was not possible to perform statistical analysis of this kinase, as the levels of total protein were too low to be quantified, p70S6K phosphorylation increased after exercise and even more when exercise was combined with Met administration.

Finally, the expression of myosin heavy chain 1/2 (MYH1/2), a muscle contractile protein, and the levels of the main MRFs were evaluated on muscle lysate.

MYH1/2 levels were higher in C+ and EX− with respect to C−, and in EX+ compared with EX− (Figure 12). The increase in levels of MYH1/2 due to Met administration was slightly increased over that induced by training in EX+.

As regards the MRFs, we observed an increase in C− compared with SED for all the myogenic factors analyzed (*t*-test, *p* < 0.0001), except for paired box 7 (PAX7) (Figure 13a, Figure 14a and Figure 15a).

Met increased myogenic factor 5 (Myf5) expression in C+ and EX+ with respect to C−, while training alone reduced Myf5 levels compared with C− (Figure 13b). Training-induced Myf5 decrease was slightly less than that of the Met effect in the EX+ group.

Regarding myoblast determination protein (MyoD), its levels were significantly higher in EX− with respect to C−. When exercise was combined with Met administration (EX+), MyoD increase was lower, although not significantly (Figure 14b).

Lastly, as regards PAX7, no statistically significant differences were observed among the groups (Figure 15b).

### 3.6. IHC Analysis of MRFs and MYH1/2

Table 3 reports the results obtained by the IHC analysis of MRFs and MYH1/2 and these confirm the data obtained by WB analysis.

In addition, IHC analysis was performed to evaluate myogenin levels in each group. Its levels appeared higher in C+ and EX− with respect to C−, and in EX+ compared with EX−, showing the same trend of expression of MYH1/2. Appendix A provide representative images of IHC analysis.

## 4. Discussion

Met is a biguanide oral antidiabetic drug for the treatment of type 2 diabetes mellitus. Its therapeutic effects are based on a combination of the improvement of peripheral uptake and utilization of glucose, a decrease in the hepatic glucose output, a decreased rate of intestinal absorption of carbohydrates, and an enhancement in insulin sensitivity without causing side effects related to hypoglycemia [1]. For these reasons, Met can be used by healthy athletes to enhance their PP during a competition, taking advantage of insulin anabolic properties without falling into hypoglycemic risk related to insulin assumption.

Firstly, we aimed to investigate the effects of Met on PP in vivo on a healthy rodent model to clarify whether this drug could be considered a doping substance.

We evaluated food consumption and body weight increase, as previous studies have reported that Met administration could have anorexiant properties and physical exercise (PE) can reduce body weight. From our results we noticed no differences among groups as regards weight gain and food consumption. We observed that Met administration increased muscle weight, both alone and combined with training, suggesting a contribution of this antidiabetic drug in muscle anabolism improvement by impacting insulin sensitivity [29]. PE did not affect the muscle weight/body weight ratio in a statistically significant way, in agreement with the previous observations that endurance exercise training, unlike resistance exercise training, does not elicit changes in size and contractile properties of muscle fibers [30]. To evaluate how training and Met administration affected rats’ PP, animals underwent the GEET at the beginning and at the end of the training period. Comparisons between the two GEETs showed that the PP of untrained rats worsened over time. Conversely, the performance of trained animals improved in the same time interval, as expected. Met administration counteracted the worsening of PP observed in untrained rats, although not in a statistically significant way, but it did not improve the positive effects of PE on PP. These results demonstrate that Met administration does not affect endurance exercise effects on PP and, consequently, Met cannot be of interest to athletes who want to ameliorate their performance. Because Met alone improved performance, even if slightly, this drug could represent an alternative to PE in subjects unable to perform it.

Then, we analyzed factors considered markers of muscle injury, such as ALT, AST, LDH and CK–MB. ALT blood levels were reported to increase after muscle damage associated with acute exercise [31]. Interestingly, we noticed an increase in ALT levels in C− compared with SED, and we hypothesize that this augment could result from muscle damage that occurred during the GEET performed 48 h before sample harvesting. Moreover, the increase in ALT levels was also evident in the Met treated group (C+), whereas in the exercised groups (EX− and EX+) these values were lower. This result suggests that training can protect skeletal muscle against damage that occurs after a bout of endurance exercise. Similar results were obtained for CK–MB levels. High blood levels of CK–MB are associated with myocardial injury and some studies have also reported peak values of CK–MB in endurance athletes [32,33]. Levels of this kinase were higher in C− compared with SED, although not to statistically significant extent, and decreased after exercise training; however, the greater decrease was observed when Met administration was combined. Just like ALT and CK–MB, AST and circulating LDH levels also increase after acute exercise, and their rise is likely due to their release from damaged muscle cells [34]. Met administration reduced levels of AST and LDH. Altogether these results suggest a role for this drug in protecting skeletal muscle against the detrimental effects of a bout of endurance exercise. Indeed, although data show that exercise protects muscles against injuries caused by an intense exercise, when it is combined with Met administration the protective effects are greater.

Skeletal muscle is actively involved in the synthesis and secretion of myokines. In particular, muscle contraction during physical activity plays an important role in promoting the release of myokines, which can exert autocrine, paracrine and/or endocrine effects to regulate skeletal muscle growth [35,36,37]. In our study we analyzed the levels of myokines in serum and skeletal muscle samples, with the aim to point out potential differences among groups related to physical training, Met administration and Met–training combination. Met administration induced an increase in FGF21, irisin, and SPARC serum levels, and a decrease in FKN in muscle. Both training and Met administration elicited a decrease in serum osteocrin levels. FGF21 is an endocrine hormone, produced in various peripheral tissues, including skeletal muscle [38]. Several studies suggest that FGF21 could be an excellent molecule to treat type 2 diabetes and could also be involved in the metabolic enhancement induced by some antidiabetic drugs [39]. Specifically, it has been demonstrated that Met-induced inhibition of the mitochondrial respiratory chain increases FGF21 expression through an activated transcription factor 4–dependent mechanism [40,41]. From our results, drug administration increased serum FGF21 levels, independently of physical training. Moreover, FGF21 levels did not increase in EX− rats, suggesting that moderate training does not influence the circulating levels of this myokine. FGF21 levels in skeletal muscle were under-determinable, confirming that it does not accumulate in the tissue where it is produced as it is rapidly secreted. In addition, the results obtained regarding FGF21 support the inducing effect of Met on the myokine reported in [40] and confirm the efficacy of the drug administration modality adopted in our experimental protocol.

Irisin is a polypeptide hormone released into the blood after PE. The increase in its circulating levels occurs from a stimulation of PPAR-γ and PGC-1α production, which results in irisin increase as a product of its precursor fibronectin type III domain-containing protein 5 (FNDC5) [39]. Irisin influences glucose metabolism in skeletal muscle and acts as a mediator of PE effects on adipose tissue metabolism, by converting white adipose tissue into brown adipose tissue [42,43,44]. Moreover, irisin decreases body weight and insulin resistance. Plasma irisin levels increase both after endurance and resistance exercise, with a higher secretion after aerobic exercise [37,44]. From our analysis, statistical differences regarding irisin levels only emerged in serum samples, and we noticed that drug administration meaningfully increased irisin serum levels. It has been demonstrated that Met increases intramuscular irisin precursor (FNDC5) mRNA/protein expression and blood irisin levels in mice [44]. Because irisin levels have been reported to also increase after PE even as our data do not highlight any differences between EX− and C−, we can hypothesize that the intensity of the training protocol used in our study was not high enough to unmask this effect. Our results confirm the inducing role of Met on this myokine and, once again, demonstrate the effectiveness of the drug administration protocol.

Osteocrin is a peptide primarily expressed in skeletal muscle, and it has been demonstrated to act as an augment in type 2 diabetic mice. Indeed, blood and muscle levels of osteocrin are associated with fasting blood glucose levels and insulin resistance index in type 2 diabetes [45]. The mechanism responsible for the decrease of osteocrin serum levels observed both in C+ and EX− is not completely clear. We can hypothesize that Met administration and physical training also support insulin activity in healthy conditions. As such, osteocrin levels could be lower in rats undergoing Met treatment and/or training. Because the decrease of osteocrin detected in EX+ was similar to the decreases measured in C+ and EX−, we can infer that training and Met affect osteocrin serum levels through the same mechanism, i.e., by supporting insulin activity.

SPARC is a multifunctional myokine expressed in satellite cells and muscle fibers that may promote myoblast differentiation. Moreover, it can be released into the bloodstream by skeletal muscle after aerobic exercise and its expression can increase when AMPK is activated [46,47]. Our results show that Met administration increased SPARC serum levels, with a further slight increase when Met was combined with physical training. Because, in muscle, AMPK is targeted by Met that induces kinase activation, we hypothesize that Met administration could lead to an increase of SPARC expression in muscles through AMPK signaling pathway and, consequently, be involved in cell differentiation.

In muscle specimens, the only myokine that showed significant differences among groups was FKN. This myokine is a chemokine produced by muscle cells and it can be both soluble and membrane anchored. It is mainly involved in muscle regeneration, with a direct effect on myogenic cells. Moreover, FKN levels increase in skeletal muscle after resistance exercise [48,49,50]. The results obtained highlight the role of Met in regulating this myokine expression in skeletal muscle. Indeed, we observed a reduction of FKN levels in rats treated with Met. We can suppose that the decrease of FKN after Met administration is a consequence of the protective effect of the drug against muscle damages induced by exercise. The regenerative process does not need to be turned on in the absence of damage, as had occurred with C−.

Subsequently, in order to achieve the other aims of the research, we focused our analysis on muscle AMPK activity and some factors involved directly and indirectly in its pathway, such as ACCß, PGC-1α, GSK3ß, AKT and mTOR.

AMPK is an energy-sensing kinase expressed in human cells, regulated by cellular energy deficit, and activated by phosphorylation in response to various cellular stresses. When intracellular AMP/ATP and ADP/ATP ratios increase, there is an enhancement in AMPK activation related to a diminished energy availability. AMPK activation is increased by short duration and high intensity exercise, whereas low intensity exercise does not always modify AMPK activity [51]. It has been shown that Met also efficiently activates AMPK in myoblasts or hepatocytes [6,52]. The comparison between SED and C− highlighted AMPK activation when untrained rats were submitted to GEET. In rats undergoing GEET and treated with Met (C+), the activation of AMPK was higher probably because Met activates the kinase by itself, as we reported above. Instead, AMPK activation was not detectable in EX−. We hypothesize that the eight-week exercise protocol has reduced the effect of GEET on muscle rather than activating AMPK. In EX+ AMPK was highly activated, but less than in C+, as if the physical training was able to dampen the effect of Met on AMPK as well as to turn off the effect of GEET.

Once activated, AMPK phosphorylates and inhibits ACCß, the muscular isoform of ACC, leading to a decrease of malonyl-CoA and a subsequent increase of fatty acid oxidation [53]. We observed that training, and even more its combination with Met administration, inhibited ACCß activity by increasing its phosphorylation. Therefore, our results suggest that both GEET and Met administration induce AMPK activation in muscle tissue. Moreover, although training is able to completely hinder the effects of GEET, and in part those of Met on AMPK, it produces a metabolic status in muscles of trained animals equivalent to that promoted by the activated-AMPK pathway, that is the inhibition of ACCß. Met administration combined with exercise supports the training effect.

The adaptation of the cellular metabolism to its energetic status induced by AMPK takes place not only through the acute modulation of key metabolic enzymes via direct phosphorylation, but also through a slower transcriptional adaptive response. PGC-1α is one of the factors by which AMPK regulates transcriptional response. It has been reported that in muscle submitted to PE, PGC-1α, once activated, translocates from the nucleus to the mitochondrial subsarcolemmal, where it is involved in mitochondrial biogenesis [54]. Accordingly, our analysis showed an increase of PGC-1α levels in muscle mitochondrial fraction of C+, EX− and EX+, supported by a similar trend of Cyt C expression levels, one of the genes under its transcriptional control. These results, together with those concerning ACCß, support the hypothesis that AMPK is not only activated by Met administration but also by training and even by their association.

AKT is a kinase defined as the most important signal transducer of insulin; indeed, its activation increases when stimulated by the hormone. AKT inhibits hepatic gluconeogenesis and enhances both glycogen synthesis and glycolysis [55]. The same effects are involved in the Met antidiabetic action that consists in counteracting peripheral tissue resistance to insulin. Our results show a significant increase of p-AKT/AKT ratio in EX+ but not in C+ and EX−. We hypothesize that an increase in AKT phosphorylation could be brought to light for a short period of time after its activation and that its detectability depends on the intensity of activation. On the basis of this hypothesis, since AKT undergoes action of both Met and training in EX+, the level of its activation, if it occurs, is certainly higher in EX+ than in C+ and EX−. Moreover, we observed an increase in p-GSK3ß/GSK3ß ratio both after treatment (C+) and exercise (EX−) and even more in EX+. This increase could result from AKT activation, as GSK3ß is phosphorylated by this kinase.

GSK3ß is a constitutively active kinase that exerts multiple actions on different signaling cascades [56]. One of its main effects is the inhibition of the glycogen synthase (GS), leading to a decreased synthesis of glycogen. Insulin counteracts the GSK3ß effect on GS through AKT activation that, in turn, inhibits GSK3ß by phosphorylating it on Ser9. It has been reported that physical training also increases the phosphorylation of GSK3ß [57]. Taken together, these data suggest that, in our model, both Met administration and exercise, though detectable only in combination, activated AKT and that this hypothesis is supported by the increase of GSK3ß phosphorylation observed after both exercise and Met administration.

mTOR activity can be influenced both by AMPK and AKT pathways, although with different outcomes [58]. Indeed, AMPK inhibits mTOR, whereas AKT determines its activation. Once activated, mTOR phosphorylates p70S6K, resulting in an increased protein synthesis. From our results, it emerged that mTOR was activated both after Met administration (C+) and exercise (EX−) and even more when Met and exercise were combined (EX+). Moreover, the phosphorylation of mTOR was accompanied by p70S6K activation in the same groups. Therefore, as both AMPK and AKT signaling pathways were activated in muscles, we assume that AKT signaling for mTOR prevails over the AMPK signaling in our model. We evaluated the activation level of mTOR by considering its phosphorylation on Ser2448, which is believed to be responsible for promoting mTOR activity [59]. Recently, it has been reported that the phosphorylation of Ser2448 leads to inactivation of mTOR as part of an inhibitory feedback mechanism induced by the activated form of p70S6K, which switches off the mTOR signal [60,61]. Despite the significance of the phosphorylation of mTOR on Ser2448, its increase is a marker of mTOR signal activation.

In support of the notion that the mTOR–p70S6K signaling pathway activation actually occurred, levels of expression of MYH1/2, a muscle structural protein related to contractile activity, increased after both Met administration and training, as well as their combination. Furthermore, we observed that the increase in MYH1/2 expression was greater after Met administration than after training.

It is well established that the levels of MRFs vary according to the activation of the satellite cells (SCs) and the different phases of the cell cycle. As it has been suggested that SCs can intervene not only in the regeneration processes but also be the source of additional myonuclei during myofiber hypertrophy and provide additional transcriptional capacity [62], we investigated the variations related to the main MRFs in our muscles.

It is known that quiescent SCs express PAX7 and Myf5 but not MyoD or myogenin. Damage to the environment surrounding SCs results in their activation (i.e., exit from the quiescent state). Proliferating SCs express MyoD and Myf5. After proliferation, when adult myoblasts begin differentiation, PAX7 is downregulated. The onset of terminal differentiation and fusion starts with the expression of myogenin, which together with MyoD activates muscle specific structural and contractile genes [63].

The analysis of MRFs expression variations in the different groups showed that both Met and PE induced the activation of the SCs. Moreover, on the basis of myogenin levels obtained by IHC analysis, Met seemed to accelerate the activation of SCs, which reached a more advanced stage of differentiation in C+, compared with Ex−. This effect was even more evident when Met was associated with PE. Myogenin levels had a similar trend to MYH1/2.

Therefore, PE and Met would determine in skeletal muscle an increase in MYH1/2 both by stimulating protein synthesis and by activating SCs. All together the results concerning the effects of Met on MRFs and MYH1/2 are in contrast with those obtained in a previous work [49], in which we studied the effects of Met on differentiating and differentiated C2C12 cells. In differentiating cells, Met induced AMPK activation but inhibited cell differentiation and arrested cell cycle progression in G2/M phase. Moreover, Met reduced MYH1/2 expression in myotubes. We speculate that the results in vitro and in vivo may differ significantly and that the role of AMPK in myogenic differentiation could be very different from that in muscle homeostasis [60,64,65,66,67].

A limitation of this work is that it was conducted with healthy animals, thus we do not know to what extent the data obtained are translatable to human subjects and this needs to be verified.

Alternatively, a strength of this work is its evaluation of the expression of most of the proteins involved in muscle exercise-induced beneficial effects and in signaling pathways that can be modulated by PE or Met.

Ultimately, taken together, our results demonstrate that both PE and Met induce improvements in muscle anabolic response to several signaling pathways, most notably the AMPK and AKT/mTOR pathways.

## 5. Conclusions

Our data highlight the way in which Met does not improve PP and that therefore it should not be considered a doping substance. Consequently, Met should not be included in the substances list banned by WADA. Moreover, Met does not alter but enhances the positive effects of PE on muscle, promoting muscle damage recovery by SC activation and the modulation of factors involved in the muscle adaptation process induced by PE.

As our results show that Met does not nullify but improve the benefits provided by physical activity on muscle, it could be used, in combination with exercise or on its own, to prevent or delay the deleterious effects of ageing and to contrast structural and functional muscle alterations.

Considering the potentially favorable effects of Met on skeletal muscle and its contractile properties, why is it that Met administration does not improve PP? As our results show that Met and exercise have additive effects on the key regulators of exercise-induced skeletal muscle adaptation, we cannot assume that Met failed to improve PP due to its negative interference in muscle adaptation to exercise. We speculate that the cause of the lack of performance gain may be the Met-induced decrease in the improvement in cardiorespiratory fitness induced by aerobic exercise reported in [18,19].

## Figures and Tables

**Figure 1 biomedicines-11-02334-f001:**
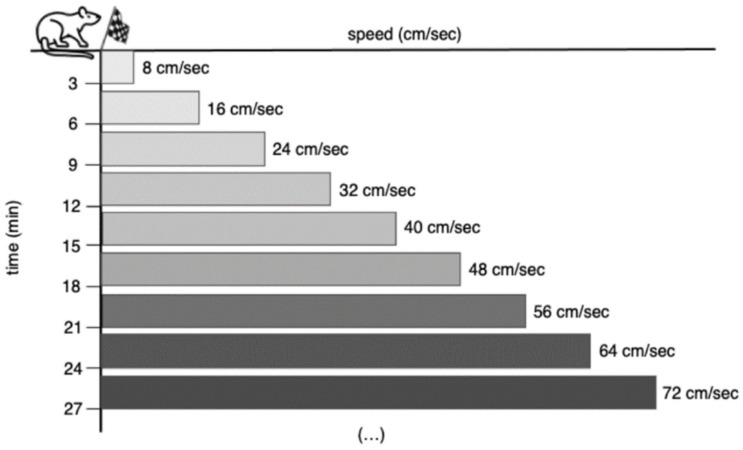
Graded endurance exercise test (GEET) protocol.

**Figure 2 biomedicines-11-02334-f002:**
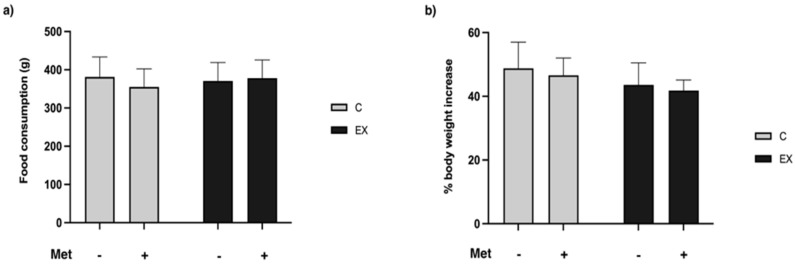
Total food consumption and body weight increase. Neither physical training, Metformin (Met) administration nor their combination affected food consumption (**a**) and body weight increase (**b**).

**Figure 3 biomedicines-11-02334-f003:**
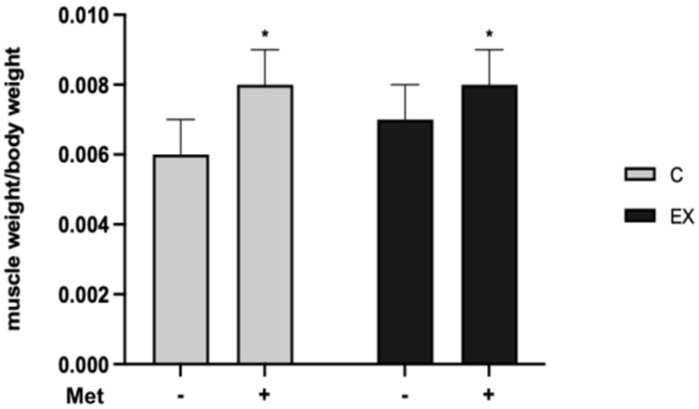
Muscle gastrocnemius weight/body weight ratio. Met administration increased the muscle weight/body weight ratio. Tukey’s test: * (*p* < 0.05) versus C−.

**Figure 4 biomedicines-11-02334-f004:**
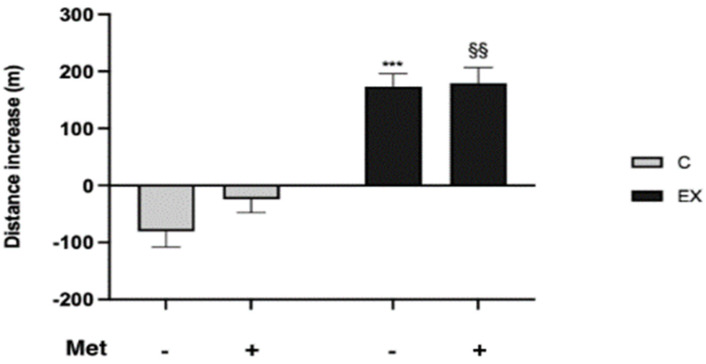
Comparison between the two GEETs: evaluation of rats’ physical performance (PP). Average increase in the distance covered by each group during the last GEET was used to evaluate the performance. Two-way ANOVA (Met administration: *F* = 7.511, *p* = 0.014; training: *F* = 406.1, *p* < 0.001 and interaction: *F* = 4.920, *p* = 0.041); Tukey’s test: *** (*p* < 0.001), versus C−; §§ (*p* < 0.01) versus C+.

**Figure 5 biomedicines-11-02334-f005:**
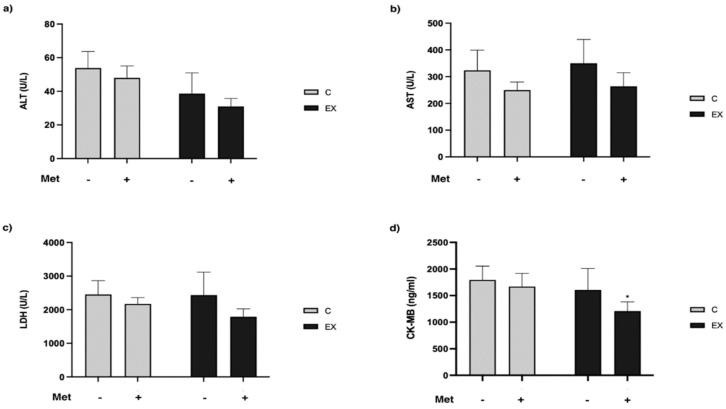
Haematochemical parameters evaluation. Blood levels of alanine aminotransferase (ALT) (**a**), aspartate aminotransferase (AST) (**b**), lactate dehydrogenase (LDH) (**c**) and creatine kinase MB (CK–MB) (**d**) in control and exercised rats are reported. PE reduced ALT and CK–MB levels; Met administration reduced AST and LDH values. Tukey’s post-hoc test did not detect significant differences between groups except for CK–MB (**d**): * (*p* < 0.05) versus C−.

**Figure 6 biomedicines-11-02334-f006:**
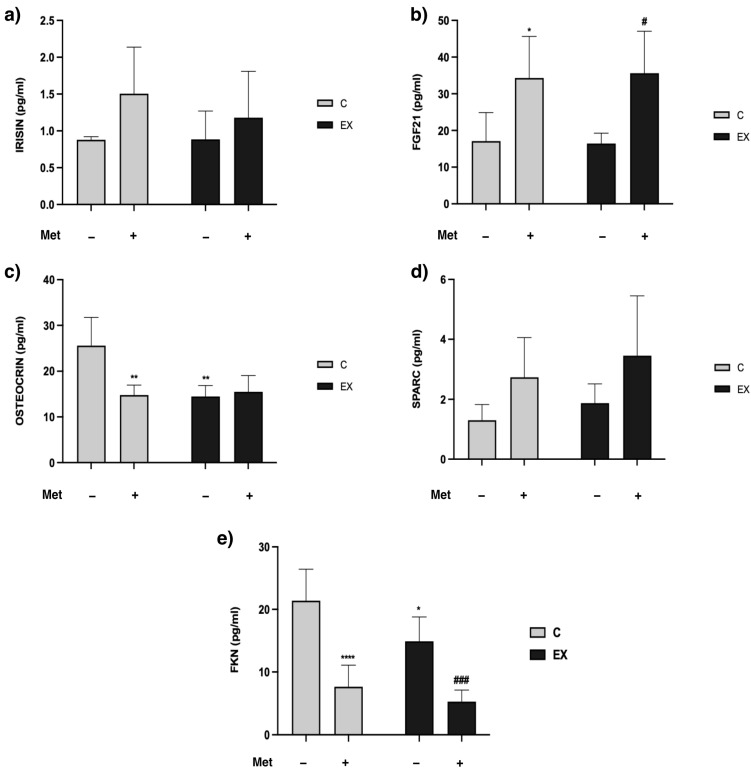
Assessment of myokines in serum and muscle. Serum levels of irisin (**a**), fibroblast growth factor 21 (FGF21) (**b**), osteocrin (**c**), secreted protein acid and cysteine rich (SPARC) (**d**), and muscle levels of fractalkine (FKN) (**e**) in control and exercised rats are reported. Met administration increased serum levels of irisin, FGF21 and SPARC. Tukey’s test: * (*p* < 0.05), ** (*p* < 0.01), **** (*p* < 0.0001) versus C−; # (*p* < 0.05), ### (*p* < 0.001) versus EX−.

**Figure 7 biomedicines-11-02334-f007:**
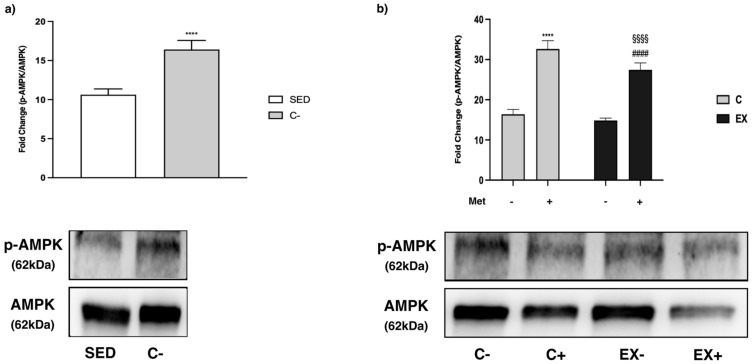
Western Blot (WB) analysis on adenosine monophosphate-activated protein kinase (AMPK) phosphorylation in muscle lysate. Representative images of WB for phospho(p)-AMPK and AMPK. Phosphorylation level is expressed as the ratio between phosphorylated and total protein. (**a**) Histograms represent band density expressed as fold change compared with SED; *t*-test: **** (*p* < 0.0001) versus SED. (**b**) Histograms represent band density expressed as fold change compared with C−. Data are representative of three independent WB analyses. Met administration increased AMPK phosphorylation. Two-way ANOVA (Met administration: *F* = 562.7, *p* < 0.0001; training: *F* = 31.03, *p* < 0.0001 and interaction: *F* = 8.991, *p* = 0.0071); Tukey’s test: **** (*p* < 0.0001) versus C−; #### (*p* < 0.0001) versus EX−; §§§§ (*p* < 0.0001) C+.

**Figure 8 biomedicines-11-02334-f008:**
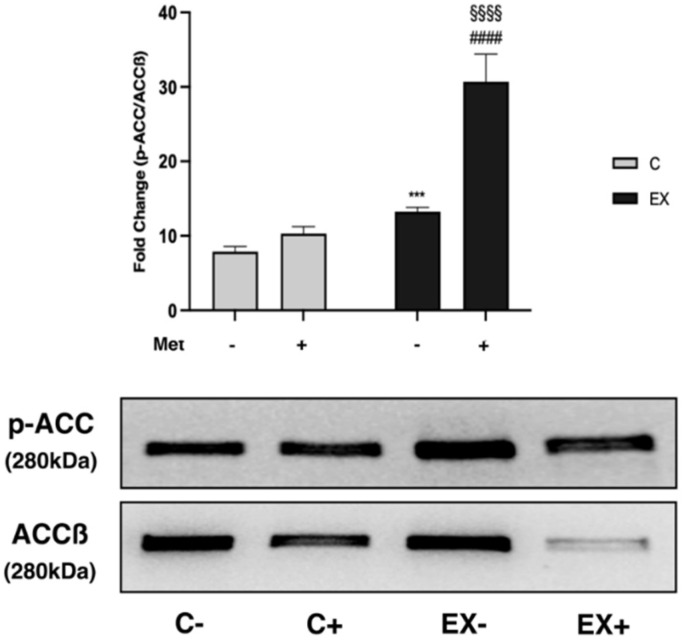
WB analysis on acetyl-CoA carboxylase beta (ACCß) phosphorylation in muscle lysate. Representative images of WB for p-ACC and ACCß. Phosphorylation level is presented as the ratio between phosphorylated and total protein. Histograms represent band densities expressed as fold change compared with C−. Data are representative of three independent WB analyses. Two-way ANOVA (Met administration: *F* = 155.1, *p* < 0.0001; training: *F* = 260.1, *p* < 0.0001 and interaction: *F* = 88.95, *p* < 0.0001); Tukey’s test: *** (*p* < 0.001) versus C−; #### (*p* < 0.0001) versus EX−; §§§§ (*p* < 0.0001) versus C+.

**Figure 9 biomedicines-11-02334-f009:**
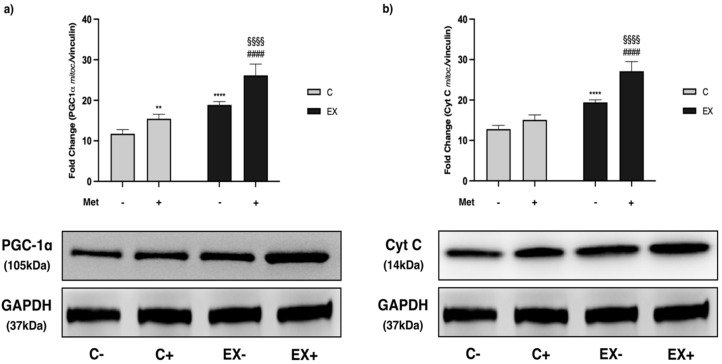
Representative images of WB for peroxisome proliferator-activated receptor gamma coactivator-1 alpha (PGC-1α) (**a**) and cytochrome C (Cyt C) (**b**) in muscle mitochondrial fraction. Histograms represent band densities expressed as fold change compared with C−. Glyceraldehyde-3-phosphate dehydrogenase (GAPDH) was used as loading control. Data are representative of three independent WBs. Two-way ANOVA (**a**): Met administration: *F* = 68.47, *p* < 0.0001; training: *F* = 179.3, *p* < 0.0001; interaction: *F* = 7.334, *p* = 0.0135; (**b**): Met administration: *F* = 72.81, *p* < 0.0001; training: *F* = 252.4, *p* < 0.0001; interaction: *F* = 21.29, *p* < 0.0002; Tukey’s test: ** (*p* < 0.01), **** (*p* < 0.0001) versus C−; §§§§ (*p* < 0.0001) versus C+; #### (*p* < 0.0001) versus EX−.

**Figure 10 biomedicines-11-02334-f010:**
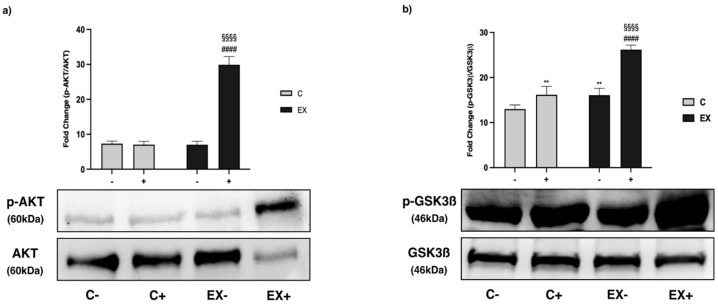
WB analysis of protein kinase B (AKT) and glycogen Synthase kinase 3 beta (GSK3ß) phosphorylation in muscle lysate. Representative images of WB for p-AKT and AKT (**a**) and p-GSK3ß and GSK3ß (**b**). Phosphorylation level is presented as the ratio between phosphorylated and total protein. Histograms represent band densities expressed as fold change compared with C−. Data are representative of three independent WBs. Two-way ANOVA (**a**): Met administration: *F* = 398.4 *p* < 0.0001, training: *F* = 85.16, *p* < 0.0001 and interaction: *F* = 418.8 *p* < 0.0001; (**b**): Met administration: *F* = 141.4 *p* < 0.0001, training: *F* = 137.3 *p* < 0.0001 and interaction: *F* = 38.31, *p* < 0.0001); Tukey’s test: ** (*p* < 0.01) versus C−; §§§§ (*p* < 0.0001) versus C+; #### (*p* < 0.0001) versus EX−.

**Figure 11 biomedicines-11-02334-f011:**
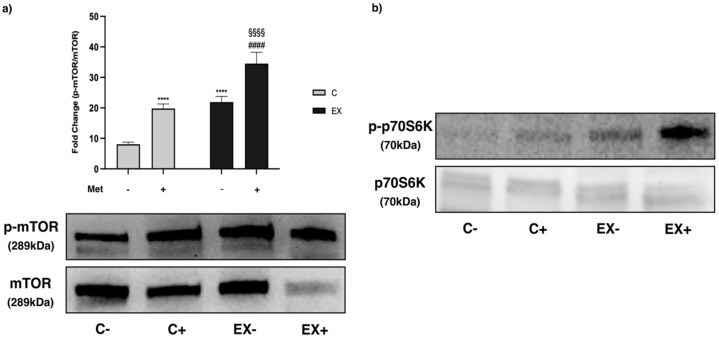
WB analysis of the mammalian target of rapamycin (mTOR) and ribosomal protein S6 kinase (p70S6K) phosphorylation in muscle lysate. Representative images of WB for p-mTOR (Ser2448) and mTOR (**a**) and p-p70S6K and p70S6K (**b**). Phosphorylation level is shown as the ratio between phosphorylated and total protein. Histograms represent band densities expressed as fold change compared with C−. Data are representative of three independent WBs. Two-way ANOVA (**a**): Met administration: *F* = 179.1 *p* < 0.0001 and training: *F* = 246.3 *p* < 0.0001; Tukey’s test: **** (*p* < 0.0001) versus C−; §§§§ (*p* < 0.0001) versus C+; #### (*p* < 0.0001) versus EX−.

**Figure 12 biomedicines-11-02334-f012:**
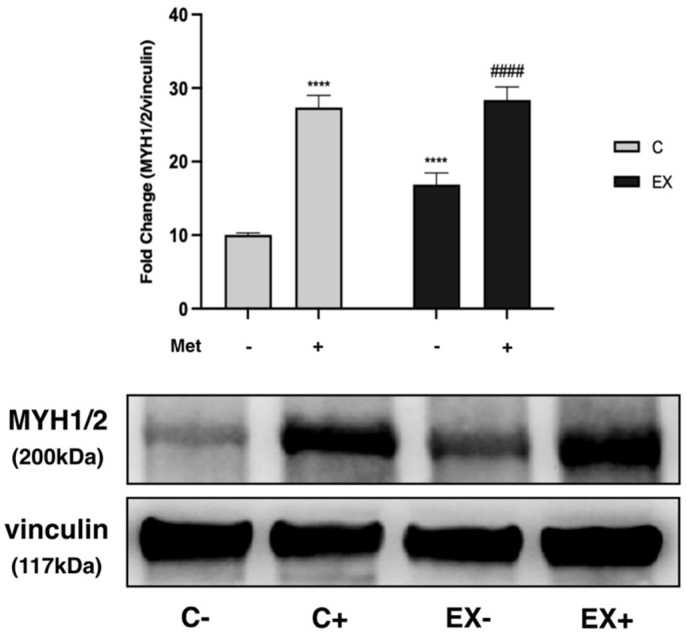
Western Blot analysis of myosin heavy chain 1/2 (MYH1/2) expression in muscle lysate. Representative images of WBs for MYH1/2. Histograms represent band densities expressed as fold change compared with C−. Vinculin was used as loading control. Data are representative of three independent WB. Two-way ANOVA: Met administration: *F* = 585.2 *p* < 0.000; training: *F* = 43.53 *p* < 0.0001; interaction: *F* = 23.90 *p* < 0.0001; Tukey’s test: **** (*p* < 0.0001) versus C−; #### (*p* < 0.0001) versus EX−.

**Figure 13 biomedicines-11-02334-f013:**
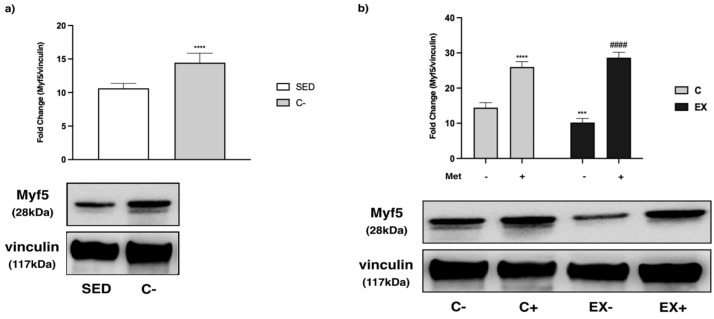
WB analysis of myogenic factor 5 (Myf5) expression in muscle lysate. Representative images of WBs for Myf5. Vinculin was used as loading control. (**a**) Histograms represent band densities expressed as fold change compared with SED; *t*-test: **** (*p* < 0.0001) versus SED. (**b**) Histograms represent band densities expressed as fold change compared with C−. Data are representative of three independent WB. Two-way ANOVA: Met administration: *F* = 673.6 *p* < 0.0001, training: *F* = 8.99, *p* = 0.0071 and interaction: *F* = 35.77 *p* < 0.0001; Tukey’s test: *** (*p* < 0.001) versus C−; **** (*p* < 0.0001) versus C−; #### (*p* < 0.0001) versus EX−.

**Figure 14 biomedicines-11-02334-f014:**
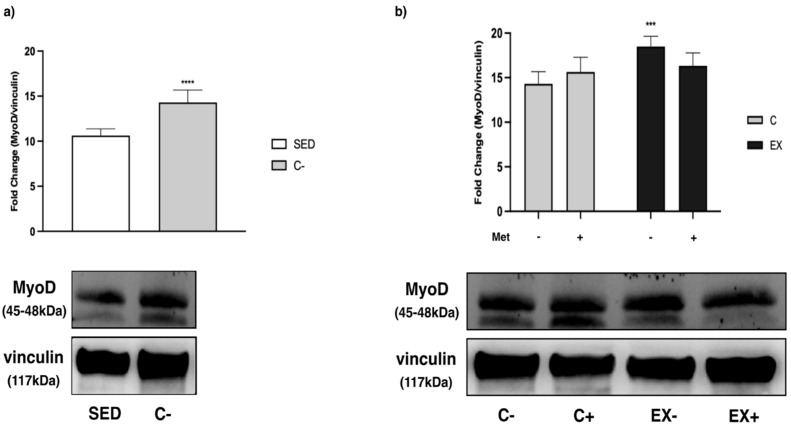
WB analysis of myoblast determination protein (MyoD) expression in muscle lysate. Representative images of WBs for MyoD. Vinculin was used as loading control. Data are representative of three independent WB. (**a**) Histograms represent band densities expressed as fold change compared with SED; *t*-test: **** (*p* < 0.0001) versus SED. (**b**) Histograms represent band densities expressed as fold change compared with C−. Two-way ANOVA: training: *F* = 17.36, *p* = 0.0005; interaction: *F* = 9.104, *p* = 0.0068; Tukey’s test: *** (*p* < 0.001) versus C−.

**Figure 15 biomedicines-11-02334-f015:**
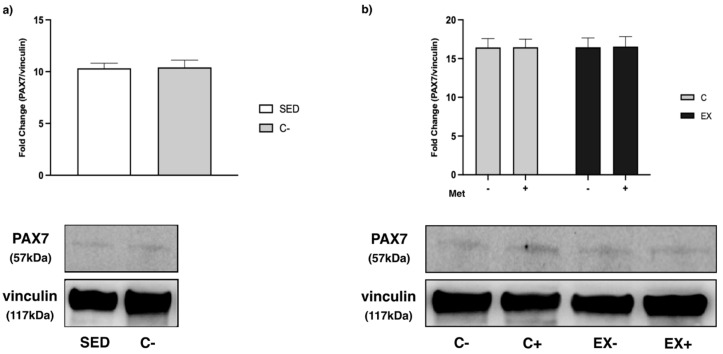
Western Blot analysis of paired box 7 (PAX7) expression in muscle lysate. Representative images of WBs for PAX7. Vinculin was used as loading control. Data are representative of three independent WBs. (**a**) Histograms represent band densities expressed as fold change compared with SED. No difference was observed between SED and C−. (**b**) Histograms represent band densities expressed as fold change compared with C−. No statistically significant differences were observed among groups.

**Table 1 biomedicines-11-02334-t001:** Exercise training protocol on treadmill.

Week	Belt Speed (cm/s)	Inclination (Degrees)	Total Time (min)
1	25	0	5
2	27	0	15
3	30	0	30
4	32	0	45
5	34	0	60
6	34	0	60
7	34	0	60
8	34	0	60

**Table 2 biomedicines-11-02334-t002:** Haematochemical parameters.

	SED	C−	C+	EX−	EX+
ALT (U/L)	35.25 ± 8.77	53.8 ± 9.88	48 ± 7.07	38.6 ± 12.36	31 ± 4.74
AST (U/L)	282.75 ± 72.07	323.6 ± 75.49	249.8 ± 29.91	350 ± 89.20	263.8 ± 50.95
Cholesterol (mg/dL)	59.5 ± 5.74	54.4 ± 8.79	61.8 ± 7.40	56 ± 5.24	58.6 ± 6.80
Creatinine (mg/dL)	0.498 ± 0.03	0.456 ± 0.02	0.450 ± 0.03	0.486 ± 0.06	0.464 ± 0.07
PCR (mg/dL)	0.01 ± 0	0.01 ± 0	0.01 ± 0	0.01 ± 0	0.01 ± 0
Glucose (mg/dL)	290.5 ± 26.96	259 ± 54.11	267 ± 47.44	308.4 ± 59.50	297.4 ± 37.27
Triglycerides (mg/dL)	118.75 ± 45.21	125.8 ± 25.62	101.4 ± 39.21	122.2 ± 51.09	87.4 ± 34.90
HDL (mg/dL)	25.5 ± 1.29	24 ± 2.74	28 ± 2.45	25.2 ± 0.84	25.2 ± 2.86
Urea (mg/dL)	45.5 ± 1.91	48.6 ± 3.13	47.8 ± 5.85	45.8 ± 5.93	50.4 ± 5.59
CK–MB (U/L)	1560 ± 359.44	1794 ± 260.35	1670 ± 250.20	1606 ± 403.34	1208 ± 174.99
LDH (U/L)	2377.5 ± 450.95	2454.2 ± 410.96	2174.2 ± 185.66	2436.2 ± 681.21	1789.4 ± 236.22

**Table 3 biomedicines-11-02334-t003:** Abundance of MRFs and MYH1/2 in muscle tissue.

	SED	C−	C+	EX−	EX+
Myf5	+/−	++	+++	+	++++
MyoD	+/−	+	++	++++	+++
PAX7	+/−	+/−	+/−	+/−	+/−
Myogenin	+/−	+	+++	++	++++
MYH1/2	+/−	+	+++	++	++++

The intensity of staining was reported as: +/−, faintly present; + present, ++, very present; +++, abundant; ++++, very abundant.

## Data Availability

All our data are reported in the manuscript.

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
