# Peer review of "Metformin Protects Rat Skeletal Muscle from Physical Exercise-Induced Injury"

_biomedicines, 2023, doi:10.3390/biomedicines11092334_

Round 1
Reviewer 1 Report
The manuscript was prepared very well. The introduction section justifies the purpose of the study. I congratulate the authors for the preparation of the manuscript
I would like to congratulate the authors for the structure of the manuscript and all the research carried out. It is highly publishable. However, there are some concerns, in part important, so the review articles need revision, see below.
Introduction
- Why is this study considered relevant?
- Why is this study necessary?
- add some of the satellite cells as muscle damage recovery elements 10.3390/proteomes10030029
Methods and Results
- It is one of the strong parts of the manuscript, these excellently described
Discussion
· Include a section on strengths / limitations.
· What mechanisms of action support these findings?
· What does this article contribute to, the authors should make their own assessment and include their own discussion of the results shown in the manuscript?
Conclusion
In the Conclusion section, state the most important outcome of your work. Do not simply summarize the points already made in the body — instead, interpret your findings at a higher level of abstraction. Show whether, or to what extent, you have succeeded in addressing the need stated in the Introduction (or objectives). 10.3390/ijms231911846
Author Response
Authors’ responses to Reviewers:
Reviewer #1
Comment 1 to the Authors: Introduction- Why is this study considered relevant?
Authors’ response/action: We accepted the reviewer’s suggestion and we included some considerations about the relevance of our study in the introduction (lines 85-89).
Comment 2 to the Authors: Introduction- Why is this study necessary?
Authors’ response/action: We accepted the reviewer’s suggestion and we explained why the study is necessary in the introduction (lines 82-89).
Comment 3 to the Authors: Introduction- add some of the satellite cells as muscle damage recovery elements 10.3390/proteomes10030029
Authors’ response/action: We thank the referee for the suggestion and we placed it in the introduction (lines 68-71).
Comment 4 to the Authors: Discussion- Include a section on strengths / limitations.
Authors’ response/action: We accepted the reviewer’s suggestion and we included limitations/strengths in discussion (lines 702-706).
Comment 5 to the Authors: Discussion: What mechanisms of action support these findings?
Authors’ response/action: We answered the question in the discussion (lines 707-709).
Comment 6 to the Authors: Discussion: What does this article contribute to, the authors should make their own assessment and include their own discussion of the results shown in the manuscript?
Authors’ response/action: We answered the question in the conclusion (lines 717-725).
Comment 7 to the Authors: Conclusion: In the Conclusion section, state the most important outcome of your work. Do not simply summarize the points already made in the body — instead, interpret your findings at a higher level of abstraction. Show whether, or to what extent, you have succeeded in addressing the need stated in the Introduction (or objectives).
Authors’ response/action: We thank the referee for the observation and we modified conclusion section (lines 712-736).
Reviewer 2 Report
This study has delved into effects of metformin on aerobic exercise, muscle mass, and major signaling pathways, markers of satellite cell differentiation, blood markers of muscle damage, and myokines. The premise was that the previous reported effect of metformin-induced insulin sensitivity might lead to metformin having insulin-like actions. Quite strikingly, metformin increased the muscle weight/body weight ratio, anabolic markers like Akt, mTOR, and S6K phosphorylation while decreasing markers of muscle damage. These results are somewhat unexpected because of metformin’s role as an AMPK activator, making the findings novel and important.
Specific comments for revision:
1. please focus the study aims on p. 2. For example, the study is not about doping substances per se; there was no aspect of aging that was addressed in the study
2. the methods on p. 3 say animals were killed 24 h after the last exercise session. Later (line 429), it’s stated that it was 48 hours post-exercise. Please clarify.
3. It would be helpful for the reader if you were to show data (if you have them) for the SED group in fig 2a/b (food, body mass). I think it would be OK to exclude the SED group from the stats, but showing the data gives context for the other groups. The same goes for the other figures if you made the measures for the SED group.
4. If there’s a significant main effect for metformin for fig 3, please report that. The same goes for other data in which it looks like there might be a main effect for metformin or exercise (figs 5, 6, 7b, 9a, 11a, 12, 13b)
5. please state in the fig 3 legend that this is data for gastrocnemius
6. please consider making fig 6 larger.
7. please enlarge the p-AMPK blots shown in 7a/b to show both the lower and the upper band for P-AMPK. Can you tell by the shape/position of the AMPK band which P-AMPK band corresponds to the AMPK band?
8. Please consider showing a better blot example in fig 8 which has roughly uniform ACCbeta bands across the groups.
9. Please state in the fig 11 legend that the mTOR phosphorylation site is S2448.
10. Please show the quantitation and stats for P-S6K/S6K shown in the fig 11b blots.
Author Response
Reviewer #2:
Comment 1 to the Authors: please focus the study aims on p. 2. For example, the study is not about doping substances per se; there was no aspect of aging that was addressed in the study
Authors’ response/action: We accepted the reviewer’s observations and we modified the aims in the introduction (lines 78, 82-83).
Comment 2 to the Authors: the methods on p. 3 say animals were killed 24 h after the last exercise session. Later (line 427), it’s stated that it was 48 hours post-exercise. Please clarify.
Authors’ response/action: We apologize if we were not clear. In line 517 (ex line 427) of the discussion, we referred to the last GEET performed 48h before the sacrifice of the animals and not to the last exercise session (performed 24h after the last GEET and 24h before the sacrifice of the animals).
Comment 3 to the Authors: It would be helpful for the reader if you were to show data (if you have them) for the SED group in fig 2a/b (food, body mass). I think it would be OK to exclude the SED group from the stats, but showing the data gives context for the other groups. The same goes for the other figures if you made the measures for the SED group.
Authors’ response/action: We showed the results obtained in the SED group only when they differed significantly from C-. We added this information in materials and methods (lines 112-113).
Comment 4 to the Authors: If there’s a significant main effect for metformin for fig 3, please report that. The same goes for other data in which it looks like there might be a main effect for metformin or exercise (figs 5, 6, 7b, 9a, 11a, 12, 13b)
Authors’ response/action: We accepted the reviewer’s observation and we included the main effect for Met or exercise in the figure legends (if necessary).
Comment 5 to the Authors: please state in the fig 3 legend that this is data for gastrocnemius
Authors’ response/action: We thank for the Reviewer’s suggestion and we specified this aspect.
Comment 6 to the Authors: please consider making fig 6 larger.
Authors’ response/action: Thank for suggestion, we made fig. 6 larger.
Comment 7 to the Authors: please enlarge the p-AMPK blots shown in 7a/b to show both the lower and the upper band for P-AMPK. Can you tell by the shape/position of the AMPK band which P-AMPK band corresponds to the AMPK band?
Authors’ response/action: Instead of enlarging the p-AMPK blot in 7a/b figure, we clipped the upper band to improve the image. However, all the original blots have been submitted as supplementary data (accordingly with Biomedicines policy). In particular, in the original p-AMPK blot (replicate 1), the highest hole on the membrane coincides with the weight of the protein and with the height of the lowest band.
Comment 8 to the Authors: Please consider showing a better blot example in fig 8 which has roughly uniform ACCbeta bands across the groups.
Authors’ response/action: We agree with the reviewer observation. Unfortunately, within all our blot replicas the image quality is similar.
Comment 9 to the Authors: Please state in the fig 11 legend that the mTOR phosphorylation site is S2448.
Authors’ response/action: We thank for the Reviewer’s suggestion and we added phosphorylation site.
Comment 10 to the Authors: Please show the quantitation and stats for P-S6K/S6K shown in the fig 11b blots.
Authors’ response/action: As reported in results section (lines 405-406) it was not possible to perform quantitation for p-p70S6K/p70S6K as the levels of total protein were too low to be quantified.